# Bacteria face trade-offs in the decomposition of complex biopolymers

**Ksenia Guseva**[1]*, **Moritz Mohrlok**[1], **Lauren Alteio**[1,2], **Hannes Schmidt**[1], **Shaul Pollak**[1], **Christina Kaiser**[1]

**1** Centre for Microbiology and Ecosystem Science, University of Vienna, Vienna, Austria, **2** FFoQSI GmbH - Austrian Competence Centre for Feed and Food Quality, Safety and innovation, Tulln, Austria

* ksenia.guseva@univie.ac.at

⊖ OPEN ACCESS

**Data Availability Statement:** Python and C++ code to reproduce simulations is available at: https://github.com/kseniaguseva/Microbial_growth_polymer_degradation and https://github.com/kseniaguseva/Enzyme_degradation.

## Abstract

Although depolymerization of complex carbohydrates is a growth-limiting bottleneck for microbial decomposers, we still lack understanding about how the production of different types of extracellular enzymes affect individual microbes and in turn the performance of whole decomposer communities. In this work we use a theoretical model to evaluate the potential trade-offs faced by microorganisms in biopolymer decomposition which arise due to the varied biochemistry of different depolymerizing enzyme classes. We specifically consider two broad classes of depolymerizing extracellular enzymes, which are widespread across microbial taxa: exo-enzymes that cleave small units from the ends of polymer chains and endo-enzymes that act at random positions generating degradation products of varied sizes. Our results demonstrate a fundamental trade-off in the production of these enzymes, which is independent of system's complexity and which appears solely from the intrinsically different temporal depolymerization dynamics. As a consequence, specialists that produce either exo- or only endo-enzymes limit their growth to high or low substrate conditions, respectively. Conversely, generalists that produce both enzymes in an optimal ratio expand their niche and benefit from the synergy between the two enzymes. Finally, our results show that, in spatially-explicit environments, consortia composed of endo- and exo-specialists can only exist under oligotrophic conditions. In summary, our analysis demonstrates that the (evolutionary or ecological) selection of a depolymerization pathway will affect microbial fitness under low or high substrate conditions, with impacts on the ecological dynamics of microbial communities. It provides a possible explanation why many polysaccharide degraders in nature show the genetic potential to produce both of these enzyme classes.

## Author summary

The decomposition of polysaccharides by microbes is a key process in the global carbon cycle. It requires the joint action of a variety of microbially-produced extracellular enzymes. They can be broadly classified into endo-enzymes, that act in the middle of polymers, and exo-enzymes, that cleave units from polymer ends. Little is known about the benefits for microbes producing a certain enzyme type and the interplay between enzyme

**Funding:** KG, LA, MM and CK have received funding from the European Research Council (ERC) under the European Union's Horizon 2020 research and innovation programme (consolidator grant agreement No 819446, granted to CK). The funders had no role in study design, data collection and analysis, decision to publish, or preparation of the manuscript.

**Competing interests:** The authors have declared that no competing interests exist.

producing strategies in mixed communities. This hampers our comprehensive understanding of decomposition in terrestrial and marine ecosystems and thus limits the prediction of decomposition processes, for example in a changing climate. Based on theoretical modelling, we revealed a fundamental trade-off in the action of these enzymes. While exo-enzymes are more efficient at high substrate conditions, endo-enzymes perform better when substrate is low. Generalists producing both enzymes expand their ecological niche of substrate availability compared to specialists only producing one of the two types. Complementary specialists only coexist in oligotrophic conditions. We conclude that producing enzymes for specific steps within polymer degradation represents relevant ecological strategies for microbes in decomposer communities.

## Introduction

The decomposition of organic matter by microbes is a rate-limiting process in the global carbon cycle [1]. Across all ecosystems, this process is governed by microbially derived extracellular enzymes, sometimes also known as eco-enzymes [1]. This is because organic matter is mostly composed of large structural polysaccharides such as cellulose or chitin, which cannot be taken directly by cells, and must be broken into smaller pieces by extracellular enzymes before utilization [2]. On the one hand, since these polysaccharides possess a diverse range of crystalline structures and linkages, one type of enzyme is often insufficient to efficiently process them. On the other hand, the production of each additional enzyme adds to the metabolic costs of microorganisms [3–6]. For challenging environments with high molecular diversity and spatio-temporal heterogeneity, such as soil, enzyme production can pose a great burden on microbial metabolism [7]. Recently, using a theoretical model Weverka et al (2023) [8] showed that chemodiversity of the organic matter pool can hinder its assimilation by microbes. An effect which would be exacerbated if microorganisms are required to produce multiple enzyme classes for each nutrient source [9]. Despite these constrains we know numerous examples of microorganisms which display genetic potential to produce a variety of enzymes involved in depolymerization and examples of enzymes with consecutive, redundant, or even synergistic actions [10–13]. Moreover, it has been shown that microbial populations go through diversification depending on the chain length, concentration and solubility of the polymer found in their surrounding environment [14]. Hehemann et al (2016) [14] showed evidence that through adaptive radiation a population of closely related bacteria can subdivide into specialized ecophysiological types adapted to the given environmental opportunities. These results showed that there is a great flexibility and variability of the depolymerization pathways and hint to the existence of trade-offs for the production of the enzymes engaged in depolymerization.

The traditional view of degradation of complex substrates assumes that at the core of this process we find a series of depolymerization steps catalyzed by glycoside hydrolases (Enzyme Commission class (EC) 3.2.1.*, e.g. for substrates such cellulose and chitin) or other lyases (e.g. for substrates such as alginate). The hydrolysis itself can proceed in two different ways: where small products are cleaved from the chain ends (effectuated by exo-enzymes); or where a polymer molecule is randomly broken at any position (as done by endo-enzymes), see Fig 1A. These enzymes appear to be redundant to a certain degree, since both have the same objective —production of small units ready for uptake. Despite its importance, a detailed inclusion of multiple steps of depolymerization process into current microbial models [15, 16] or into the models of organic matter decomposition in general [17–20] remains lacking. A notable effort

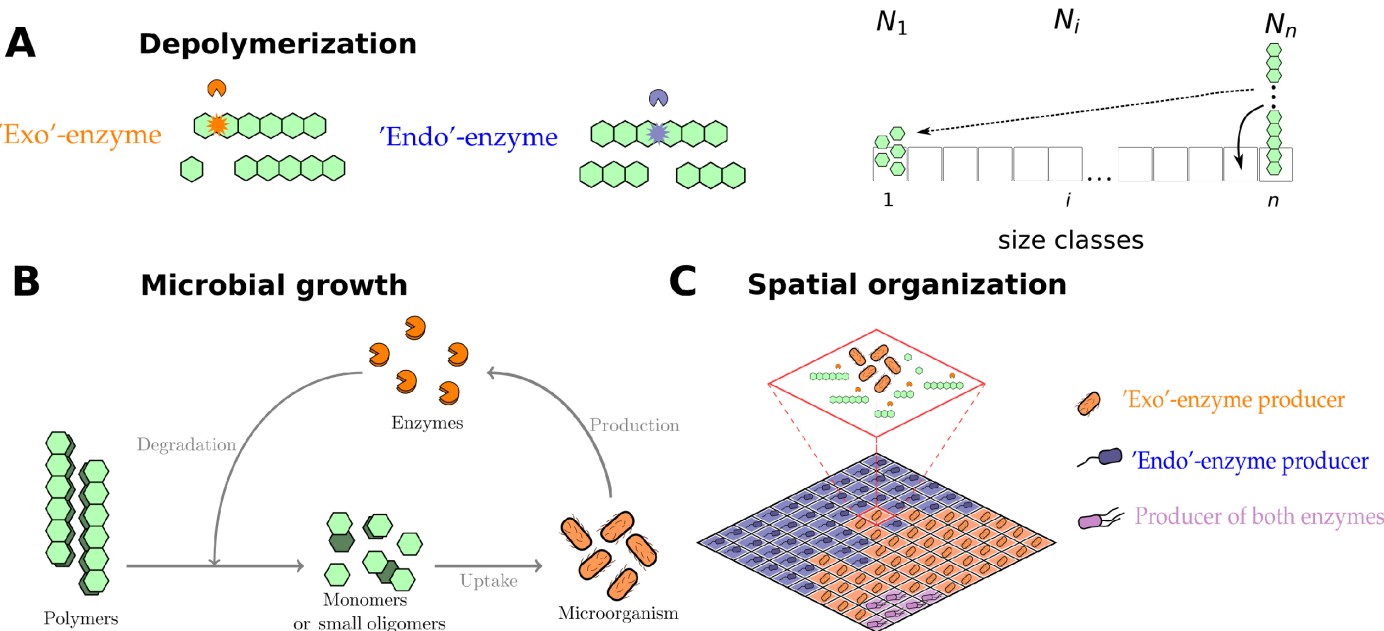

**Fig 1. Features included in the microbial degradation model.** (A) Depolymerization dynamics due to activity of exo or endo-enzymes. On the right we show the division of polysaccharides into different classes (from 1 to $n$) according to their length ($N_1$ representing the concentration of monomers—they all belong to class 1, $N_i$ represents the concentration of a oligomer of size $i$). Arrows represent possible fluxes between these size classes which result from enzymatic reactions. (B) Microbial growth dynamics, which includes depolymerization of a complex substrate, uptake of small subunits (monomers) and enzyme production. (C) Schematic representation of the spatial grid model, where each microsite represents a microhabitat with a small microbial population. This population can colonize neighboring habitats after its biomass riches a certain threshold.

in this direction was made by Sainte-Marie et al. (2021) [21], who included a continuous framework (which approximates the discrete polymer size distribution by a continuous one) to incorporate mass flow between different polymer size classes in a model of soil organic matter turnover. However, their approach still lacked an explicit inclusion of enzyme (Michaelis-Menten) kinetics and was only well-defined in the presence of very long chain molecules [22].

We propose a new framework that incorporates depolymerization dynamics into a microbial growth model that effectively addresses the aforementioned gaps [16, 23]. While our analysis is broadly applicable, we highlight the illustrative example of chitinases. Chitin degraders appear to use varied strategies for chitin degradation, where we distinguish specialists—those that exclusively produce one enzyme class (either endo- or exo-enzymes)—from generalists—those that produce both enzyme types (endo- and exo-enzymes simultaneously). To what extent and why microbes adopt specialist or generalist strategies in chitin degradation is still unclear. Our study addresses the following questions: (i) Could there be advantages of producing different enzyme classes at the same time? (ii) Which trade-offs in the production of different enzyme classes are faced by microorganisms individually or within communities? And (iii) how do these trade-offs affect microbial growth in different conditions, especially in complex environments like soils?

Our new framework allows us to approach these questions from a theoretical perspective. We assume that potential trade-offs may be different depending whether microbes live in well mixed or spatially explicit environments, so we include both in our analysis. In spatially structured systems, which host a significant portion of microbes on earth (e.g. soil) [24], the diffusive exchange of products is known to play a central role in self-organization within consortia [25–29]. To model such systems, we incorporate polymer degradation by means of population

balance equations into an individual-based spatially explicit microscale microbial community model. This approach allows us to track the spatial distribution of poly and oligosaccharide molecules of different sizes in space and time, as well as the microbes feeding on them. Using a theoretical approach, we analyse the advantages and the disadvantages of the two depolymerization types and discuss possible synergies among them. To contextualize our modelling approach we start our study by performing two surveys for chitinases: one based on genetic sequencing and another on enzyme activity assays. These two surveys help us to explore the prevalence of exo(endo)-enzymes and to understand to what extent microbes may adopt different degradation strategies in nature, complementing our theoretical analysis.

## Materials and methods

### Assessing the prevalence of different chitinase types in bacteria

With the first survey we identified the genomic potential of bacteria to produce selected chitinase classes. We screened the genomes of bacteria for selected KEGG identifiers via two tools that target: (i) soil bacteria—JGI tool [30] and (ii) all environments—Anotree [31]. These tools allowed us to obtain a list of all taxa, with their phylogenetic classification, that display genetic potential for the selected KEGG identifiers [32]. For the second survey, we performed a systematic literature search to assess the experimentally tested activity of exo- and endo-chitinases, that used 4-Methylumbelliferone (4-MUF)-marked oligomers to quantify exo-chitinase (chitobiosidase) and endo-chitinase activity colormetrically or fluormetrically. From these studies we also listed (see S1 Overview) all the species capable to degrade chitin, and the respective type of activity recorded by these assays, details in S1 Text.

### Model overview

The main components of our model are: (i) microorganisms (with their biomass given by $C_b$), (ii) enzymes produced by these microorganisms (which can be exo or endo, with concentrations $E_{\mathrm{exo}}$ and $E_{\mathrm{endo}}$ respectively), (iii) polymers of different sizes ($N_i$) and dissolved inorganic nitrogen (DIN) pool. The system evolves according to the equations:

$$C_b(t + dt) = C_b(t) + \text{growth}, \tag{1}$$

$$E_j(t + dt) = E_j(t) + fr_j \cdot \text{enzyme production} - \text{decay}, \tag{2}$$

$$N_{j>1}(t + dt) = N_{j>1}(t) - \text{degradation} + \text{degradation}, \tag{3}$$

$$N_1(t + dt) = N_1(t) + \text{degradation} - \text{uptake}, \tag{4}$$

$$\mathrm{DIN}(t + dt) = \mathrm{DIN}(t) - \text{uptake}, \tag{5}$$

where uptake depends on the microbial cell area, while growth and enzyme production depend on the stoichiometric balance between carbon and nitrogen taken up (the details are given in the S1 Text). The details of the degradation dynamics are given below (notice that only monomers can be taken up by the cells). In this work we analyse two main types of systems: one that assumes well mixed conditions and is fully deterministic; and a second where we couple this deterministic dynamics to a non-deterministic spread of population in space and a random mortality.

## Enzyme kinetics and population balance equations (PBE)

As explained previously the exo-enzymes $E_{exo}$ cleaves long polymeric chain $N_i$ into small organic molecules (monomers, $N_1$) which can be taken up by the cell, following:

$$E_{exo} + N_i \underset{k_2}{\overset{k_1}{\rightleftharpoons}} (E_{exo}N_i) \overset{k_3}{\rightarrow} E_{exo} + N_{i-1} + N_1 \qquad (6)$$

On the other hand endo-enzymes $E_{endo}$ are able to cleave any bond:

$$E_{endo} + N_{i+j} \underset{k_2}{\overset{k_1}{\rightleftharpoons}} \left( E_{endo}N_{i+j} \right) \overset{k_3}{\rightarrow} E_{endo} + N_i + N_j \qquad (7)$$

We model the degradation of chitin by means of population balance equations (PBE), following the framework proposed by Suga et al (1975) [33]. This approach divides the available polymers in groups depending on their size (monomrs, dimers, trimers, n-mers, etc), we denote the number of polymers of size $i$ as $N_i$, see Fig 1A. The number of polymers of each size class evolves, $\frac{dN_i}{dt}$ following the fragmentation dynamic established by the enzyme which catalyzes the reaction, see S1 Text. This dynamics decreases the numbers of polymers of one class and correspondingly adds to other size classes. One important property of the system is that the total amount of monomers in the system is conserved $\sum_{i=1}^{n} iN_i = M$. We use Michaelis-Menten kinetics for both enzyme types; the reaction rate $v_{exo_i}$ of exo-enzyme is given by

$$v_{exo_i} = k_{exo} \frac{E_{exo}N_i}{k_m + \sum_{j=1}^{n} N_j}; \qquad (8)$$

and the reaction rate for the endo-enzyme is:

$$v_{endo_i} = k_{endo} \frac{(i-1)E_{endo}N_i}{k_m + \sum_{j=1}^{n}(j-1)N_j}, \qquad (9)$$

in both equations the $k_m$ is the half saturation rate of both enzymes, which is set to 0.5 nmol/ mm$^3$. The turnover number (catalytic constant) of the two enzymes are given by $k_{exo}$ and $k_{endo}$. For exo-enzymes $k_{exo}$ is fixed to 0.82 nmol nmol C$^{-1}$ h$^{-1}$. Conversely the turnover number of endo-enzymes ($k_{endo}$) can be similar or lower than of exo-enzymes depending on the settings. The full set of equations used can be found in the S1 Text.

## Microbial growth dynamics

Microorganisms are modeled following the dynamics established in [16, 23]. In this model microorganisms grow by taking up monomers, released trough the degradation of large polysacharides by extracellular enzymes, produced by these microorganisms (the general framework is presented in Fig 1B). This approach has the advantage of explicitly including the stoichiometric C and N fluxes. The microbial metabolism is divided into uptake, maintenance, enzyme production and growth. Microorganisms are able to uptake only the smallest substrate units, in our case those are monomers, although these settings can be extended to include other sizes as well. Additional nitrogen is taken up from dissolved inorganic nitrogen pool. A fraction of carbon from the uptake is used for maintenance. Another fraction of carbon together with nitrogen is used for enzyme production (CN$_e$, fixed C:N ratio of enzymes). The rest is used for growth (CN$_m$ fixed C:N ratio characteristic to microbes). Each metabolism step, such as growth and enzyme production, includes an additional fraction of respired carbon. If there are leftovers of carbon or nitrogen those are respired or eliminated, respectively. The eliminated nitrogen re-enters the DIN pool. In the absence of nutrients in the

environment the microbial biomass decreases due to a requirement of maintenance. We set up a minimum biomass per microsite (0.08 mol C), after which the microorganism is considered dead. The enzymes, on the other hand, have a fixed lifetime, established by their decay rate $r_e$. See S1 Text for additional details and parameters.

### Spatially explicit dynamics

The spatial model is also an extension of [16, 23], see Fig 1. The spatial model consists of a $100 \times 100$ spatial grid representing a 1 mm$^2$ soil area. Each microsite represented in this grid is considered to be 10 $\mu$m$^3$. Each grid cell may contain: a single type of microorganism, enzymes, substrate (in the form of polymers of all sizes) and dissolved inorganic nitrogen. It is important to emphasize that we consider that everything is well mixed within a single microsite. Enforcing the condition of only one type of microorganism per microsite allows us to model consortia in which each microorganism type has preferential access to the degradation products of the enzyme it produces, see Fig 1C.

Microorganisms are not allowed to move but can colonize the neighboring sites by division/reproduction. When microbial biomass within a microsite exceeds a certain limit, the biomass is divided and part of it is transferred to the neighboring site. Although for simplicity we assume that biomass is divided in half, it is important to recognize that altering the division method may lead to slight changes in the dynamics of population spreading. If all the surrounding sites are occupied the division does not occur. Enzymes are also transferred together with microorganisms; we assume that they are attached to the cell wall and do not diffuse away. Random death events are also included in the spatially explicit model. Finally, polymers of size larger than 10, are insoluble and do not diffuse, contrary to all smaller sizes that can diffuse. We consider that the diffusion rate of each n-mer depends on its size, taking into account Rouse-Zimm relation for polymer diffusion [34] we can write the diffusion of an oligomer of size $i$ as $D_i = D_0(i^{-1/x})$ with $x = 1$, where $D_0$ is the diffusion rate of the monomer. All parameters can be found in the S1 Text.

### Model scenarios

We analyse four model scenarios: (i) degradation dynamics by a fixed concentration of enzymes; (ii) we compare the growth of microorganisms specialized in producing either exo- or endo-enzymes in different conditions (amounts of substrate); (iii) then we analyse the growth of generalists that invest in both enzymes depending on the proportion of each enzyme produced; (iv) finally we analyse which conditions can harbor consortia of specialists.

## Results

### Assessing the prevalence of different chitinase types in bacteria

Before we engage with the main model analysis, we show the results of two surveys that illustrate the prevalence of different strategies in polymer degrading microorganisms: one based on genetic sequences and another on enzyme activity assays, for details see Materials and methods and S1 Text. In these surveys we were looking for indicators to differentiate specialists—producers exclusively of one enzyme class (in particular of only endo or exo-enzymes), from generalists—producers of multiple enzyme types (in particular both endo and exo-enzymes together). Although a large diversity of carbohydrate-active enzymes (CAZymes) are known, the exact functions of most of them have never been tested biochemically. Chitinases, particularly endo- and exo-chitinases, are an informative test-case. Endo-chitinases are known to cleave any bond, while the exo-chitinases act by cutting off dimers from the ends of the

polymeric chain. Monomers are then generated through the activity of an additional enzyme N-acetylglucosaminidase (also known as chitobiase) that specifically cleave dimers into monomers [35].

The first survey has shown us the variety of enzymes involved in chitin degradation, but it also demonstrated that predicting endo- or exo-activity from genomes is extremely challenging. The enzymes are usually classified according to their sequence diversity and evolutionary origin which only partially overlap with specific activity types. Taking our example of microbial chitinases, they belong to three major glycoside hydrolases (GH) families: 18, 19 and 20 [10, 11, 36]. Functional classifications, on the other hand, e.g., Enzyme Commission (EC) classes or KEGG identifiers, are still evolving and have ambiguous information, e.g. K01183 (mainly GH18, with endo and exo-activity), K01207 (chitobiase, GH20), and K03791 (mainly GH19, that display mostly but not exclusively endo activity). Our survey shows that chitobiase (N-acetylglucosaminidase) is widespread in bacteria (Fig 2A) and predominantly can be the only type of chitinase that a bacteria has. Conversely, the other two types of sequences were found in distinct but overlapping groups (Fig 2B). Notably K03791 (GH19) was mostly found in Proteobacteria and was absent in Firmicutes. Although this survey illustrates that microorganisms exhibit a plethora of diverse chitinase combinations, it still fails to provide a proper comparison of prevalence of exo and endo-activity.

To complement our sequence-based overview we performed a literature review of experimentally characterized microorganisms where endo/exo-chitinase activity has been distinguished using custom-designed fluorogenic substrates (see details in Materials and methods, S1 Text and S1 Overview). We find again that large part of chitin degraders seem not to specialize in a single enzyme class of chitinases, showing both exo- and endo-activity types (Fig 2C).

## Degradation by endo-enzymes is characterized by a nonlinear monomer release rate

In our first simulation scenario we aimed to characterize the activity of enzymes without the presence of microorganisms. For that we introduced a pool of substrate and a small, but fixed, concentration of endo or exo-enzymes and recorded the degradation dynamics. Our results show that the main difference between the two enzyme types during degradation is the rate of monomer release. In the absence of microbes, exo-enzymes show a constant rate of monomer release (Fig 3A and 3B). This rate depends on the number of polymer chains, with larger number of chains leading to faster degradation (Fig 3B). Note that for exo-enzymes the number of polymer chains never changes and as degradation proceeds, with each monomer cleaved out, each one of these chains becomes shorter and shorter. For the endo-enzymes, on the other hand, the release rate is not constant: the process starts with only a few monomers produced, but this number increases nonlinearly with time. Moreover, the duration of this initial monomer production lag depends on the concentrations of polymers, with high concentrations leading to fewer monomers released in the starting period (Fig 3C). Despite having this initial delay, the endo-enzymes are much more efficient in the presence of long chains, as we show in Fig 3A, where we show that for a chain size of $n = 200$ the total time necessary to degrade the total polymer pool $T_{deg}$ under endo-activity is significantly shorter. This difference in $T_{deg}$ between endo and exo-enzymes decreases for small chains, see Fig D in S1 Text. In summary, the model shows that specialists producing exo or endo-enzymes would experience different monomer release rates. Next we discuss how these differences are reflected on the microbial growth of such specialists.

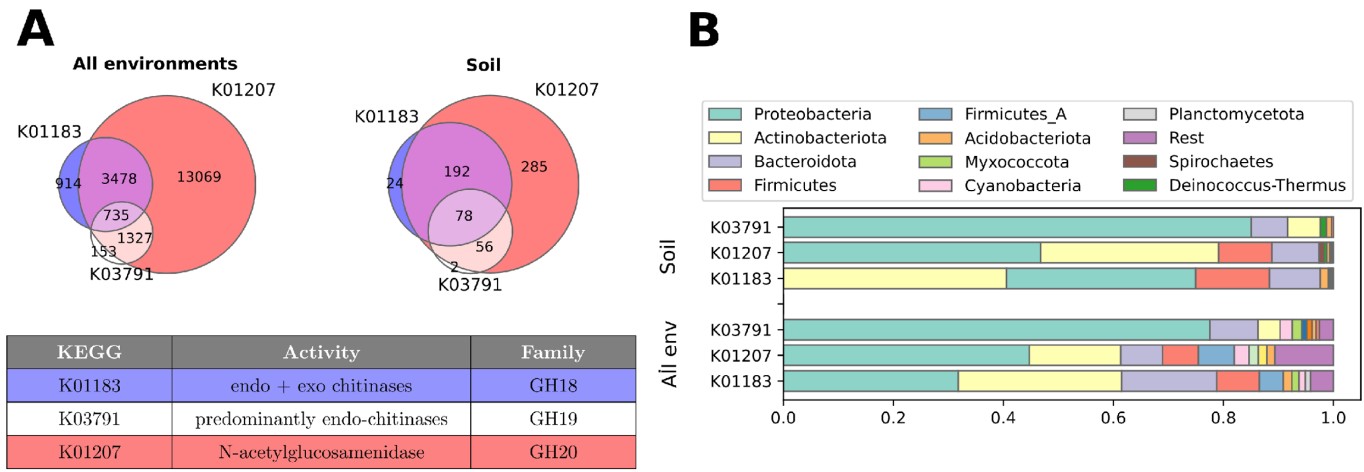

| KEGG | Activity | Family |
|---|---|---|
| K01183 | endo + exo chitinases | GH18 |
| K03791 | predominantly endo-chitinases | GH19 |
| K01207 | N-acetylglucosamenidase | GH20 |

**Fig 2. Mode of action and distribution of different classes of chitinases.** (A, B) Results of a genetic survey on prevalence of different chitinase classes (KEGG identifiers indicated) within different microbiomes (using Anotree tool—to screen all environments, using JGI—for soils only). (C) Recordings of exo or endo-chitinase activity in chitin degrading microorganisms, the data is based on a literature survey of studies which classified enzyme activity based on reactions with fluorogenic oligomers (details in the S1 Text). We indicate the number of strains used for the classification (total appearances), and the origin of the bacteria (e.g. marine, soil, carnivorous plant etc).

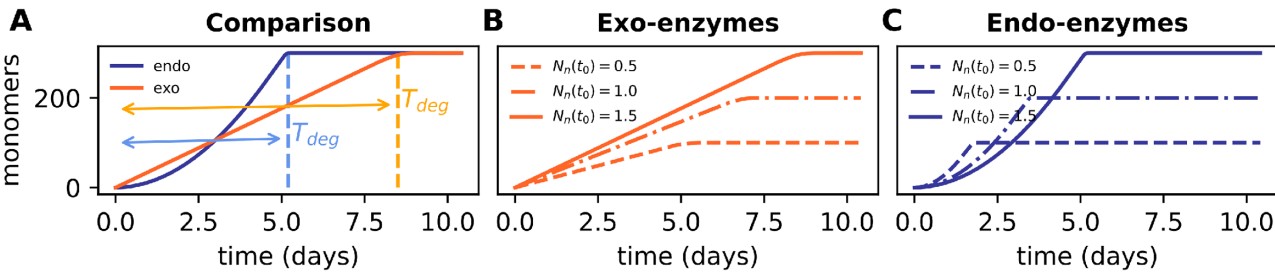

**Fig 3. Differences in the dynamics of release of monomers by exo and endo-enzymes.** (A) Comparison between depolymerization dynamics of exo and endo-enzymes, the system starts with M = 300 nmol/mm$^3$ of monomers all connected into chains of size n = 200 (i.e. the concentration of polymers in the initial pool is $N_n(t_0)$ = 1.5 nmol/mm$^3$). $T_{deg}$ indicates the time until the available pool of substrate is fully degraded. Note $T_{deg}$ is shorter for endo-enzymes. (B) Monomer release by exo- (in concentration $C_e$ = 3 nmol C/mm$^3$) and (C) endo-enzymes ($C_e$ = 3 nmol C/mm$^3$), with same turnover rates $k_{exo} = k_{endo}$ = 0.82 nmol nmol C$^{-1}$ h$^{-1}$, for different initial concentrations of polymers (in nmol /mm$^3$) of size n = 200, note the differences in the final amount of monomers obtained. The solid lines in (A, B, C) represent the monomer release dynamics for identical initial pools.

## Specializing in exo or endo-enzyme production decreases the ecological niche of microorganisms

Although having both endo and exo-enzyme types may appear redundant, since they are simply different ways to degrade the same substrate, our results show that specializing in one or the other enzyme production can lead to significant disadvantages. To understand them, we added microbial growth dynamics to the enzyme model. First we analyze the microbial growth in well mixed conditions. We initialize the population with a small amount of enzymes (in an analogous way the population can be initialized with a small amount of monomers). This is necessary to trigger enzyme production within the population which shortly after becomes either self-sustained or dies out [37].

In well mixed conditions exo-producers grow well only in high substrate concentrations (Fig 4A). For them the quantity of polymer chains, rather than their size, becomes the limiting factor for nutrient uptake. If the uptake rate is insufficient for the maintenance and further enzyme production of microorganisms, they will not survive, which is evident in the low-substrate conditions (Fig 4A and 4B). Endo-producers face opposite challenges and cannot grow in high substrate concentrations. While environments rich with polymers potentially offer enough nutrients, these chains must first be broken down to monomers for the uptake, a process which usually takes multiple steps for endo-enzymes. As we have shown in the previous subsection larger nutrient amounts result in longer delays before sufficient monomers appear to sustain endo-enzyme producers (Fig 3C). This delay stems from the fact that endo-enzymes cleave any bond within a polymer with equal probability. Consequently, there are numerous pathways to break down large molecules, most of which do not result in monomer production. In other words, the probability of generating monomers from long chains is extremely low when using endo-enzymes. However, as polymers gradually decrease in size as a result of degradation, the probability of generating monomers gradually rises. In essence, the reaction requires time to break down the polymers into progressively smaller chains, ultimately leading to the production of monomers, see lower panel of Fig 4B. If this delay is too long, the microorganisms cannot grow under those conditions, leading to population collapse (Fig 4A and 4B). In other words there is an intrinsic time period during which the microbes are able to wait for the appearance of monomers. If they do not get this return on time the population is unable to survive.

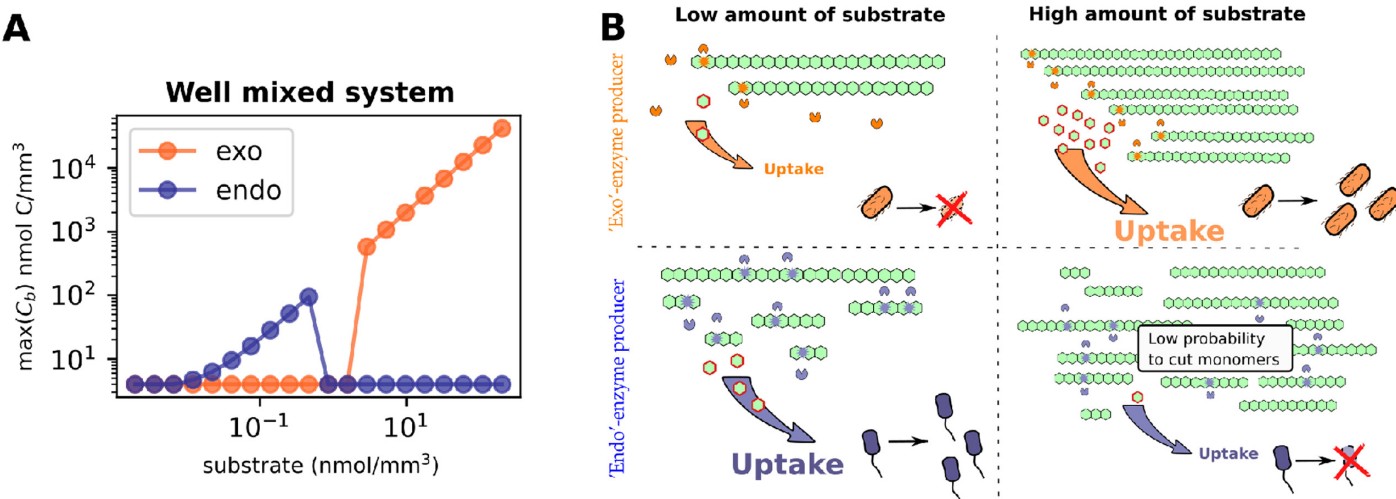

**Fig 4. Growth conditions of endo and exo-enzyme producers.** We show results for degradation of polysaccharide chains of size $n$ = 500 by microorganisms in different set-ups. For comparison, the turnover rates of the two enzymes are the same, $k_{exo} = k_{endo} = 0.82$ h$^{-1}$ (A) Maximum biomass (max($(C_b(t)$) achieved during degradation of different pools of polymers in well mixed conditions. The results show that high substrate amounts hinder the growth of endo-producers and promote the growth of exo-producers. (B) Schematic representation of trade-offs faced by microorganisms. Exo-enzymes can release as many monomers as there are chains available for them to cut, as a consequence exo-enzyme producing microbes starve in the presence of few chains and survive in high substrate conditions. In contrast, endo-enzymes rapidly process low substrate amounts, releasing a high number of monomers. However, in high substrate conditions, since the probability to cut any chain bond is the same, they primarily cleave available chains into oligomers, which cannot be taken up by cells. This can lead to a prolonged absence of monomers, potentially resulting in starvation among endo-enzyme producing microbes.

Most microbial communities do not experience well-mixed conditions, and must deal with the challenges imposed by spatial structure. In this context, an important constrain is the loss of breakdown products due to diffusion. We measured the extent of this loss and its effect on growth by using a spatially explicit model, in which we tracked the biomass of a single microsite (i.e. a grid cell), with microorganisms that are unable to divide and spread to neighboring sites but oligomers which can diffuse away (Fig 5A and 5B). We observed that while exo-producers are only mildly affected, endo-producers lose nearly half of their potential biomass even when diffusion is weak (Fig 5A and 5B). To be able to understand if this loss may affect the spatial self-organization of the microbial community we followed with further spatial analysis, now allowing microbial reproduction and spatial spread of the population. We have chosen a diffusion rate (30 $\mu$m$^2$/h) that allows endo-producers to also accumulate sufficient biomass for reproduction to occur. In the spatial context, endo-producers are still limited to substrate-poor environments, while exo-producers thrive in conditions with high polymer availability (Fig 5D and 5C respectively). Additionally, the varying substrate levels have an impact on the spatial organization pattern. Since endo-producers have limited growth, they expand in a narrow ring as the population consumes all the nutrients during habitat expansion (Fig 5D). On the other hand, exo-producers take longer to deplete all the available resources and move forward before the microsite is fully depleted (Fig 5C). Our findings also indicate a significant delay before endo-producers develop into a sizable population, and their spread through the grid is considerably slower.

## Advantages of being a generalist: Niche expansion and synergy

Our results suggest that microorganisms might benefit from producing both endo- and exo-enzymes simultaneously. To assess whether this is the case, we examine the effects of varying

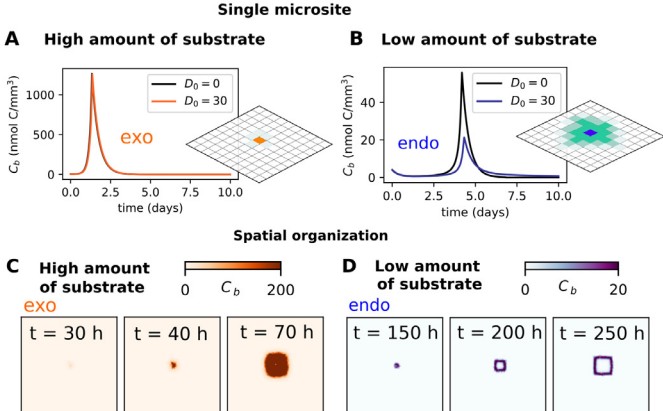

**Fig 5. Differences between endo and exo-enzyme producers: How much products they lose to diffusion and how they organize in space.** As before we show the results for degradation of polysaccharide chains of size $n = 500$ by microorganisms in different set-ups. The turnover rates of the two enzymes are the same, $k_{exo} = k_{endo} = 0.82$ h$^{-1}$ (A, B) Time evolution of biomass $C_b(t)$ in a single microsite of a spatially explicit model. We assume conditions where microorganisms are unable to divide, but oligomers up to size 10 (see Methods) are able to diffuse. We compare the settings with no diffusion ($D_0 = 0$) and with diffusion ($D_0 = 30\mu$m$^2$/h). (C, D) Snapshots of the spatial distribution of microbes (biomass concentration $C_b(x, y, t)$) at indicated times. High substrate concentration: M = 1500 nmol/mm$^3$; Low substrate concentration: M = 150 nmol/mm$^3$.

the proportions of produced endo ($1—fr_{exo}$) and exo-enzyme ($fr_{exo}$) fractions in microbes able to produce both enzymes. As before, first we address the catalytic efficiency of a constant pool with these two enzymes, which we uncouple from microbial growth. Fig 6 compares the degradation efficiency, evaluated trough the degradation time $T_{deg}$ necessary to degrade the whole substrate pool into monomers, considering different enzyme fractions. In a process where the two types of enzymes have the same turnover rates, the endo-enzymes finish to degrade the pool of polymers faster, with smaller $T_{deg}$ (Fig 6A), especially when longer chains are present. When considering the more characteristic scenario where endo-enzymes have longer intrinsic turnover rates than exo-enzymes a different picture emerges. In such cases, combining endo and exo-enzymes can result in a synergistic effect that speeds up degradation, particularly when the two enzymes exist in a specific proportion ($fr_{exo} \sim 0.8$, Fig 6B). Note that this effect is prominent only for the case when monomer pool is trapped into longer chains and the exo-enzyme activity is limited by the availability of polymer ends (see $n = 500$ in Fig 6B).

Our results show that having the ability to produce the two enzyme classes also allows microorganisms to grow at an expanded range of nutrient concentrations compared to enzyme specialists (Fig 6C). The proportion in which exo and endo-enzymes are produced is also important in this context, as we show in Fig 6C.

Finally, diffusion of breakdown products away from microorganisms can also affect the performance of generalists. We quantify this loss by comparing the total biomass of microbes in a single microsite after growth with no diffusion ($\max(C_b^*)$) to the biomass in the presence of diffusion ($\max(C_b)$, Fig 6D). Our simulations reveal that carbon loss increases dramatically with the fraction of endo-chitinase produced, in both high and low nutrient conditions. Overall our simulations reveal that producing both enzymes classes can result in synergistic effects and a broader niche for generalists. However this faster degradation dynamics compared to pure exo-enzyme producers also comes at the cost of increased diffusion-mediated carbon loss.

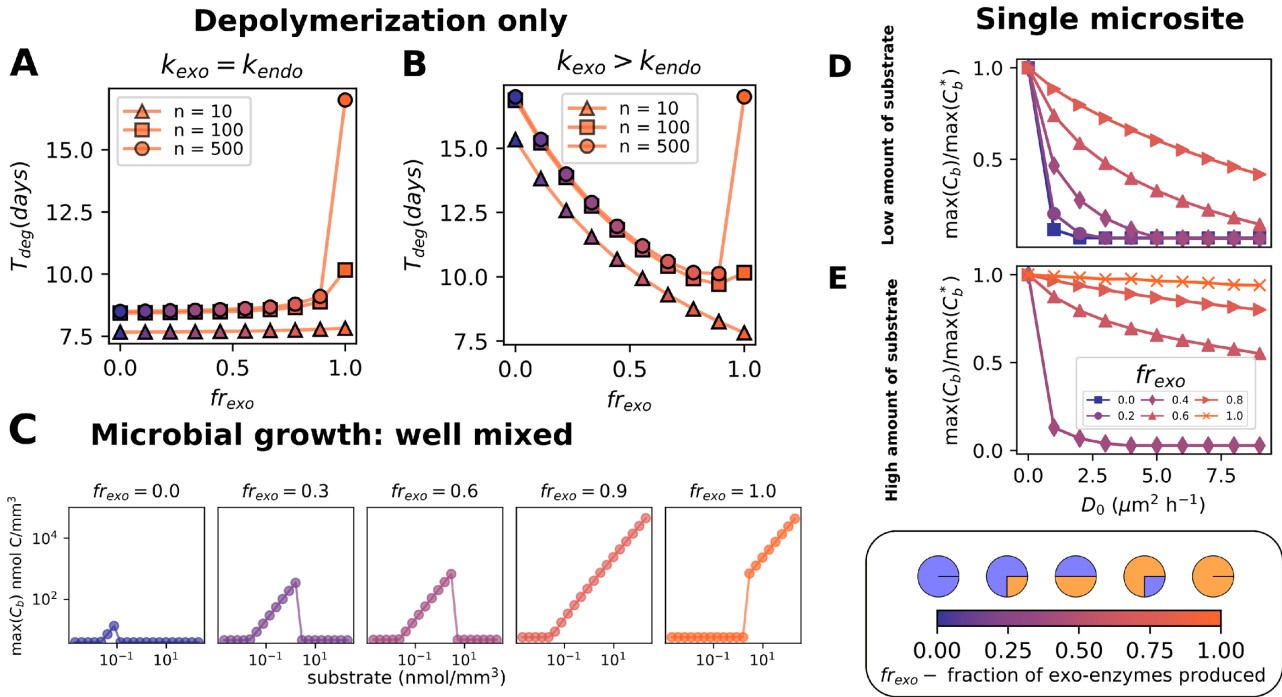

**Fig 6. Advantages of generalists (producers of both endo and exo-enzymes).** (A, B) Comparison of the total degradation time ($T_{deg}$) of different proportions of exo and endo-enzymes acting simultaneously. The simulation represents only the depolymerization dynamics in the absence of microbes in homogeneous conditions. The complex substrate pool was initiated with a total of 500 nmol/mm³ monomers distributed between chains of n-mers (n indicated in the figure labels). A pool with 3 nmol C/mm³ of two enzymes in different proportions act on the substrate (x-axis and color gradient with purple indicating endo-enzymes and orange indicating exo-enzymes), while the turnover rate or exo-enzyme is $k_{exo} = 0.82$ h$^{-1}$, the turnover of endo-enzyme is (A) $k_{endo} = k_{exo}$ and (B) $k_{endo} = k_{exo}/2$. (C) Maximum biomass reached from different initial concentrations of polymers for microorganisms producing different fractions of endo and exo-enzymes. The parameters are the same as the ones used in simulation in Fig 5, except for the turnover of endo-enzymes, which is $k_{endo} = k_{exo}/2$. (D, E) Comparison of the maximum growth in the absence of diffusion max($C_b^*$) to the maximum growth in the presence of diffusion max($C_b$), for different diffusion coefficients $D_0$ for (D) low, M = 150 nmol/mm³, and (E) high substrate, M = 1500 nmol/mm³. Each curve represents a different proportion of exo to endo-enzymes ($fr_{exo}$). Note that for low diffusion the ratios of maximum possible biomass would be the same, and with an increase of diffusion the growth decreases due to loss of soluble oligomers.

## Conditions for the formation of a consortium of endo and exo producers

In the previous section we considered a single generalist organism producing both enzyme types. A population composed of both exo- and endo-enzyme producing specialists will have both enzyme types, analogous to a generalist organism. We used simulations to study whether the two specialist types can co-exist, and the influence of mixing types on population dynamics. In the case of unequal turnover rates of the two enzymes, as in the previous section the coexistence of the two taxa (of exo and endo producers) is not possible. The faster turnover rate for exo producers gives them an uneven advantage, and they rapidly dominate in all conditions. When the turnover rates are the same, however, coexistence is possible in low substrate conditions (Fig 7B and 7C). We find that the two types spatially segregate but still influence each other. The growth of the consortium is significantly faster compared to pure endo-specialists. In conditions with high substrate amounts, the consortium is quickly dominated by exo-producers, which share little of produced monomers resulting in the extinction of endo-producers (Fig 7A).

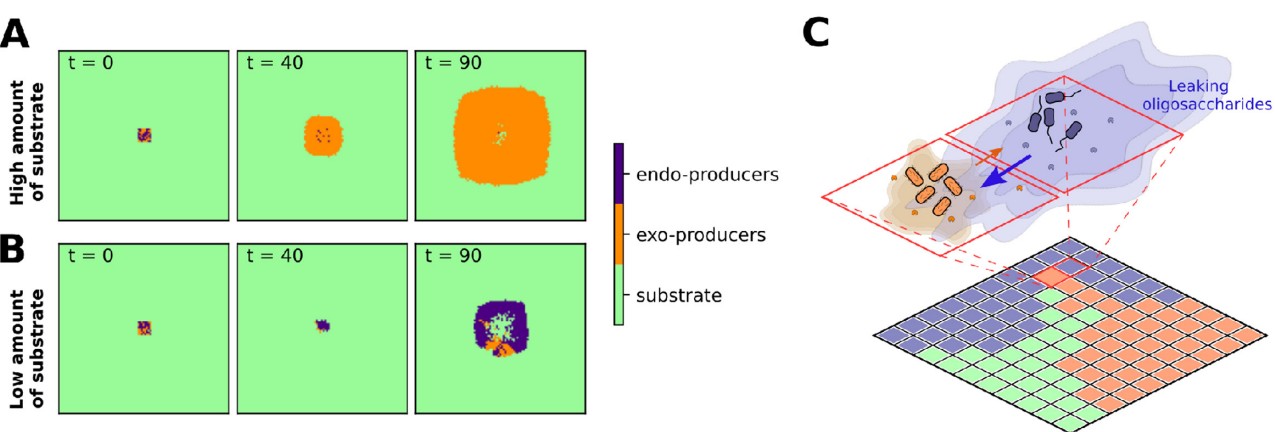

**Fig 7. Time evolution of consortia of endo and exo producers**: (A) in conditions with high substrate amount initiated with polymer concentration of 1500 nmol/mm³ and (B) with low substrate amount initiated with 150 nmol/mm³. The initial carbon is distributed in chains of size $n = 500$. Cooperation is only possible when the turnover rates of the two enzyme types are comparable, $k_{exo} = k_{endo} = 0.82$ h$^{-1}$. (C) Schematic representation of two interactions within colonies trough the diffusion of degradation products (note that each microsite can be occupied by a single type of microorganism). While endo-producers share more oligomers with their neighbors, exo-producers consume most of the degradation products within a single microsite.

## Discussion

We have presented a novel framework to model growth of heterotrophic microbes in the presence of large polymers, which explicitly accounts for the depolymerization dynamics. Our modeling results demonstrate that microbial growth depends not only on the amounts of carbon present in the system but also on the form in which this carbon is found, specifically the polymer chain size. We describe the key features of the degradation by exo and endo-enzyme types and the trade-offs faced by microorganisms specializing in the production of a specific enzyme class. Our model shows that specialization constrains growth potential. Moreover, our results demonstrate that these disadvantages can be overcome by generalists, that produce both enzymes in a specific ratio. All trade-offs are summarized in Table 1. These findigs can be contrasted to our two surveys that illustrate the variability of degradation strategies in nature. While the first one showed the limitation of predicting endo or exo activity from genomic information, the second one showed the prevalence of generalists (producers of both enyzmes) among chitin degrading microorganisms. Our model findings are in agreement with this overwhelming presence of generalists among chitin degraders, see Fig 2 or examples in [38–40], and thus offer a possible explanation for this observations.

Our work has revealed intrinsic trade-offs with deep consequences for microbial community assembly and organic matter decomposition neglected by previous model-based studies. Current models that target microbial growth [15, 16, 41, 42] or organic matter turnover [17, 20], oversimplify degradation of complex substrates by assuming that the outcome of each reaction between enzymes and polysaccharides necessarily produces units ready for uptake. This indirectly implies kinetics which can only result from exo-enzyme activity. We show that the choice of depolymerization pathways leads to fundamental differences for the microbial activity at the small scale, with impact on organization of microbial communities. Notably, the implication of these trade-offs for large scales are still to be uncovered.

At the center of our findings are the contrasting requirements for growth of exo- and endo-enzyme producers. While exo-enzyme producers need a large number of polymer molecules, endo-producers find rich conditions challenging due to a concentration-dependent delay in

**Table 1. Advantages and downsides of alternative strategies of microorganisms in degradation of complex substrates.**

| Type | Pros | Contras |
|---|---|---|
| exo-producers | 1. Fast initial growth in substrate rich conditions.<br>2. Smaller fraction of monomers loss to diffusion.<br>3. Can expand the niche by growing with endo-producers. | 1. Cannot grow in low concentrations of long polysaccharides.<br>2. Degradation time and uptake rate directly depend on the number of polymeric chains. |
| endo-producers | 1. Can grow in low concentrations of long polysaccharides.<br>2. Shorter total degradation times for comparable turnover rates. | 1. Cannot start growth in substrate rich conditions.<br>2. A lag in the initial growth with increase of polymer abundance.<br>3. Very leaky function.<br>4. Enzymes may have slower turnover rates.<br>5. It has difficulty to expand the niche by growing with exo producers. |
| generalists | 1. Synergy between endo and exo-enzymes speeds up degradation of long chains.<br>2. Grows in low and high amounts of substrate. | 1. Subjected to diffusive loss in spatial settings. |

their ability to produce monomers. In summary, our results show that different types of environments would favor the producers of one or the other enzyme: oligotrophic environments favoring endo-producers and carbon-rich ones favoring exo-producers. They also suggest that in a succession dynamics, the exo-producers could establish the initial population in an environment rich with complex substrate, followed by endo-producers when the substrate becomes depleted. Note that exo and endo-producers analyzed in our work, apart their investment into different enzymes, share identical metabolic strategies (i.e. they invest in the same way in resource acquisition and growth). Moreover, the two types would perform similarly in the presence of simple substrates (monomers, dimers or other small oligomers) and the difference is only observable in the presence of complex substrates.

The spatial simulations show additional restrictions for endo-producers, which are subjected to higher losses of intermediate products (diffusing oligomers). This is because depolymerization dynamics of endo-enzymes produces a high number of intermediate products, these oligomers are also subjected to diffusion but cannot yet be taken up by the cell. Consequently, individual endo-producers experience a reduced return on their enzyme investment, "leaking" large part of the processed resources to the surrounding environment. For this reason in spatial settings such microorganisms should be much more affected by costs of endo-enzyme production. On one hand we can speculate that while this loss penalizes individuals, it may favor larger colonies, as leaking of enzymatic breakdown products to the surrounding environment would go to the benefit of neighboring microbes in the same colony. On the other hand all these members of the colony have to compete with additional processes for the uptake of small organic molecules. One of such processes in soils is the adsorption of diffusing molecules on mineral surfaces or other forms of immobilization of organic material, for example by aggregate formation [43]. Additional direct competitors for the uptake are members of other species, "cheaters", who do not degrade the polymer themselves, but are known to "steal" degradation products away from enzyme producers for their own growth [23, 42, 44, 45]. Their proliferation will be favored by such loss. Conversely, exo-producers are able to take up the monomers almost at the rate of monomer production, sharing less with their surroundings and consequently cooperate less within colonies. For this reason they would be far less

exploited by cheaters. Although we have not explored the cheater-producer feedbacks in our study, such interactions between individuals exploiting the system and those contributing to "public goods" have been extensively analyzed in theoretical models [42] and documented experiments [45, 46], revealing major consequences for organic matter build-up in soils [23]. In summary our results indicate significant differences for spatial-dynamics and organization of exo and endo-producers. These differences are observable in the amount and sizes of diffusing breakdown products and therefore are reflected on potential cellular uptake within consortia. Such effects will have a profound impact on the ecological dynamics, since the diffusion and distribution of soluble byproducts and their complexity were empirically linked to social interactions and spatial self-organization of microorganisms [47, 48], as well as densities of formed colonies and levels of cooperation and competition within them [49, 50]. It is important to note that our model considers the dynamics at very small spatial (the grid cell is 10 $\mu$m$^3$) and relatively short time scales. The diffusive exchange we model, both between and within micro-habitats, is tied to connectivity of the soil's water film. The number and size of aqueous bacterial habitats would change with the soil water content, which would follow the drought or rainy periods. Finally, we emphasize that our model results for specialists rely on the assumption that the enzymes and polymers are relatively well mixed within a single microsite, however small scale (within microsite) spatial organization mixing of extracellular enzymes and polymers may also affect the speed of the degradation process.

For generalists, i.e., microbes that produce both enzymes, we observe a broader niche, suggesting a strong advantage of having both enzyme classes present in the system. We numerically show the synergy between the two enzymes in agreement with previously elaborated conceptual models [12, 13]. Synergies between endo and exo enzymes, where well documented for example for endo and exo-chitinases from *Serratia marcescen* [51]. Moreover, we see that this advantage only appears when most of the resources are used to produce exo-enzymes, and only a small fraction goes into endo-production. Despite these two significant advantages (synergy and niche expansion), in spatial settings the generalists also "leak" larger fractions of products to their surroundings when compared to exo-producers. This offers certain disadvantages, and we see in nature examples to overcome this issue. Microorganisms such as *Vibrio* spp., for example, have transport systems that can uptake chitin oligomers. The molecular and physiological details of chitin degradation is well documented for these microorganisms and they have endo and exo chitinases within the periplasm and outside the cell. They are known to have large chitoporins that transport molecules with sizes up to 6 NAGs into the periplasm, where additional enzymes proceed with degradation [52, 53]. Since the diffusion rate decreases with oligomer size, it is clearly advantageous to minimize the losses by taking up not only the final but also intermediate products, preventing them to diffuse away.

Since each one of these enzymes are costly and perform "leaky" functions, they constitute potential targets for gene loss [54, 55]. According to the black queen hypothesis (BQH) [56, 57], within a community the evolution would favor individuals that lose complex metabolic pathways that produce public goods, as long as these public goods continue to be generated by other community members. This could be the reason for the great variability in the endo/exo-activity pattern for some marine microorganisms such as *Proteobacteria* (e.g. see species from the genera *Pseudoalteromonas*), which we observe in our literature overview (Fig 2C). Our simulation results, however, suggest that the enzyme losses of endo or exo-enzymes in microbial communities would have different effects depending on the environmental conditions. We observed that consortia combining endo and exo-producers are only stable in low substrate conditions since in high substrate exo-producers have high advantages due to fast growth rate and low product loss (lower diffusive losses). Our theoretical findings hint that if microorganisms are subjected to variable environmental conditions which is

common in nature [58], with changing substrate concentrations, the organisms that lose exo-enzymes would be more likely to survive than the ones that lose endo-enzymes. As stated previously, however, our overview on enzyme activity (Fig 2C) suggests that both losses can occur and we can observe microorganisms with endo-activity only. This is the case for the strains of marine bacteria such *Labrenzia alba*, which apparently exhibits only endo-enzyme activity. If we eliminate the possibility that the absence of exo-activity is linked to cultivation conditions or the employed assay, this would imply that the specific strain must genuinely exhibit oligo-trophic characteristics, thriving exclusively in environments with limited substrate. Other such examples of specialization we see in bacteria found associated to carnivorous plant. We can only speculate that these microorganisms inhabit environments that offer alternative nutrient sources that enable endo-producers to endure while awaiting for an increase in their uptake rates. Moreover, the aquatic settings could facilitate mixing, which attenuates the disadvantages of such specialists within consortia.

In general, from our review of enzyme activity we see that we have very limited knowledge on the endo and exo-classes of microbial chitinases. Moreover there is still a large number of "unclassified" or uncharacterized sequences when performing genome analysis. Some of these could be associated with chitinases or be alternative chitinases, but have not been functionally characterized or confirmed. We have shown that the different pathways in the degradation dynamics can have fundamentally distinct outcomes for microbes. In summary, considering that bacteria have a finite amount of resources, investing in one enzyme necessarily reduces the investment available for the other. In this context our model analysis demonstrates an intrinsic trade-off: that enzyme-producing bacteria can excel either at low substrate concentrations (by producing endo-enzymes) or at high substrate concentrations (by producing exo-enzymes), driven by the inherent depolymerization dynamics of the two types of enzymes. As a consequence we show that bacteria would have to optimize the two ratios to survive in both conditions. Finally, our theoretical analysis illustrated that microbial consortia can form in the degradation of a single complex substrate, by splitting the production of apparently similar but complementary enzymes. We also showed how interactions within such consortia can change from positive to negative depending on substrate availability. Such context dependent interactions broaden our understanding of the dynamic nature of microbial communities and can be used to expand our microbial population models (Lotka-Voltera or Resource-Consumer models) to account for such effects [59, 60]. In summary numerical models can shed light on the constrains and trade-offs to which microorganisms are subjected, offering theories about evolutionary strategies of microorganisms and also possible interactions among them.

## Supporting information

**S1 Text. Supplementary material.** Detailed model description and additional results. (PDF)

**S1 Overview. Literature overview of chitinase activity.** A systematic literature search to assess the experimentally tested prevalence of exo- and endo-chitinases in microorganisms. (CSV)

## Author Contributions

**Conceptualization:** Ksenia Guseva.

**Data curation:** Moritz Mohrlok, Hannes Schmidt.

**Formal analysis:** Ksenia Guseva.

**Funding acquisition:** Christina Kaiser.

**Investigation:** Moritz Mohrlok, Lauren Alteio, Hannes Schmidt, Shaul Pollak, Christina Kaiser.

**Methodology:** Ksenia Guseva, Christina Kaiser.

**Software:** Ksenia Guseva.

**Supervision:** Christina Kaiser.

**Writing – original draft:** Ksenia Guseva.

**Writing – review & editing:** Ksenia Guseva, Moritz Mohrlok, Lauren Alteio, Hannes Schmidt, Shaul Pollak, Christina Kaiser.

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
