## [Decision Letter · Decision Letter 0]

15 Dec 2023

Dear Dr Guseva,

Thank you very much for submitting your manuscript "Bacteria face trade-offs in the decomposition of complex biopolymers" for consideration at PLOS Computational Biology.

As with all papers reviewed by the journal, your manuscript was reviewed by members of the editorial board and by several independent reviewers. In light of the reviews (below this email), we would like to invite the resubmission of a significantly-revised version that takes into account the reviewers' comments.

Please make sure that you convincingly addresses all issues raised by both reviewers. The issues they raise will certainly require significant revisions of the main text, including additions to clarify several points and to better place the results in the context of the relevant literature. 

We cannot make any decision about publication until we have seen the revised manuscript and your response to the reviewers' comments. Your revised manuscript is also likely to be sent to reviewers for further evaluation.

Sincerely,

Tobias Bollenbach

Academic Editor

PLOS Computational Biology

Natalia Komarova

Section Editor

PLOS Computational Biology

We can only consider a revised version if it convincingly addresses all issues raised by both reviewers. The main issues they raise will certainly require major revisions of the main text, including additions to clarify several points and to better place the results in the context of the relevant literature. Please also consider their additional suggestions.

Reviewer's Responses to Questions

**Comments to the Authors:**

Reviewer #1: Review by Stefano Manzoni, Stockholm University

The manuscript by Guseva and co-authors investigates advantages and disadvantages of producing different types of enzymes in microbial populations and communities. This topic is timely as we strive to capture the decomposition process in more mechanistic ways in mathematical models. The proposed model setup is sound and the results are interesting. Despite the complexity of the model and interpretation of some of the results, the model and methods in general are explained in a very clear way. Figures provide an information-rich—but still accessible—overview of the findings. I also liked the mix of literature review to set the stage and context, and theoretical developments. It is an unusual combination, but I found it effective. I have some comments that might help linking the current study with other recent ones, and some suggestions on the model description. Detailed suggestions are listed at the end.

Main comments

1) Novelty and link to recent literature. The mathematical model proposed here is quite novel, but recent work has tackled similar questions and should be considered to place the advances and results into context. Weverka et al. (2023, https://doi.org/10.1016/j.soilbio.2023.109161) investigated trade-offs in enzyme production from the point of view of enzyme allocation and costs. Sainte-Marie et al. (2021, https://doi.org/10.1038/s41467-021-21079-6) have already integrated endo- and exo-enzymes in their decomposition model. They also consider a range of substrates, while you focus on more or less complex carbohydrates. What are the differences between your approach and theirs? What are the methodological novelties of this work?

2) Fate of depolymerization products. L267-268: diffusion-driven losses are actual losses only with respect to a microbe/microbial colony, but they might be gains for microbes in other grid cells (as noted in L330). Advection is a much more powerful mechanism for removal of dissolved organic matter. But there are other mechanisms that would act a bit like diffusion—removing depolymerization products from the vicinity of microbes. For example, depolymerization products can be stabilized on soil minerals or trapped into stable aggregates. While the kinetics of these processes are not dependent only on solute concentrations, but also on availability of mineral surfaces/dynamics of aggregates, I would still discuss their potential role in the context of your findings.

3) Model description. Please see below in the detailed comments some suggestions. In particular, I would suggest double-checking units and reporting units (and definitions) for all symbols in a separate table.

Detailed comments

L21: what are the time scales involved in this example from the literature? are they comparable to time scales relevant in this work?

L37: missing section reference

L163: perhaps “that can diffuse”?

L170: “to characterize”

L183: could you clarify why high concentrations lead to fewer monomers. Is that because endo-enzymes are busy breaking down large polymers and need time to make their way to monomers?

L221: do you mean “of each microsite”?

Figure 5: some fonts are too small to read. Perhaps move panels C-D to a second row of panels to allow for larger fonts and plots

L335: “to take up monomers”

L350: this seems a model assumption more than a hypothesis

L364: just a detail: “spp.” not in italic

L370: “by taking up”

L390: another small detail: I might use “exhibits” rather than “demonstrates”

Supplementary information:

- Enzyme dynamics are described by a difference equation, while polymer mass balances are described by ordinary differential equations. I would homogenize the presentation of the equations, and explain how they are solved numerically (if using a finite difference scheme, were results numerically stable?).

- Please check the SI for typos (e.g., “Considering that the a microsite” after Eq. 11; “of cells/h^-1” in Table S5; fr in symbol efr should be subscript). Also check units; e.g., in Eq. 11 units and dimensions are mixed (mol/time instead of mol/hour). There are some unnecessary brackets in some formulas (e.g., Eq. 6; definition of V), but that is a very minor detail.

- I would suggest adding a table with all symbols defined and reporting their units. It would help understand equations that seem to have inconsistent units, like Eq. 9 where units should be mol/h per cell but the units on the right-hand side seem to be (interpreting the explanations in the text…) mol per channel per hour times channel area per cell. Channel number and channel area don’t have same units, so I could not recover the correct units of mol/h per cell.

Reviewer #2: The main claim of the manuscript is that a trade-off emerges from biopolymer decomposition due to the differential dynamics of depolymerizing enzymes, or in other words, different depolymerization mechanisms have different ecological consequences. The originality of this finding, as far as I know, is quite high, and it has significant consequences to understand microbial ecological dynamics. Among others, I highlight the theoretical possibility of formation of consortiums under one single complex substrate, which expands the traditional view in microbial ecological dynamics that is usually examined in the light of Gause’s (competitive exclusion) principle. This is just one example, and therefore, this manuscript has potentially general relevance in the field of microbial ecology.

However, I have some main comments and a few minor comments about your manuscript, which you can found below.

Introduction.

The detour that you take on the introduction from l. 34 and so on… well, I think you can easily lose readers there. The three paragraphs starting in line 34 have parts of introduction, methods, and even results. In my opinion, the different bits of information have to go to the relevant sections of the manuscript. I advise to state clearly the questions that you try to answer with the two surveys and add the background information in the methods (except the relevant information for the rest of computational approaches you followed, or the general discussion). I believe that you selected chitinases to inspire your computational research, so probably it would be better to introduce these three paragraphs after the mention to the gap of inclusion into models of the depolymerization process.

Model.

I think that more detail should be included about the model. Without looking at references or the supplementary information, I can not decide if your model is deterministic/stochastic or in continuous/discrete time, neither I can see a clear description of the state variables. In my opinion, this makes less likely the reproducibility of this work. Please include more details in the main text.

Additionally, I find that using reaction-based schemes to describe models, they are easier to understand, so I'd include it in the main text for all mechanisms in the model. For example, it is unclear to me whether your endo-enzymes are able to produce monomers. In fact, there's a mismatch between Fig. 1A and Fig. 2A, as they don't represent the same set of reactions. If endo-chitinases only produce dimers, and the chitobiase is needed for the absorption of monomers, then chitobiase should be modelled explicitly. The same can be said for exo-chitinases. An additional step in the metabolism may well change the dynamics of microbial growth.

Moreover, when I arrived to line 220 or so, I realize that in the methods there is no specifications on what are you going to look in with your, and you just introduce it in the results. I encourage you to make all your analyses explicit in the methods.

Finally, some minor things. I think there is a need to justify why do you allow only a single type in your grid cells, and why do you divide biomass by half. Does your results change in you keep change the proportion of biomass in the new cell vs the original one?

Use of the term trade-off.

I am not so sure that microbes are facing a trade-off here, as generalist are predominant. What is the trade-off here? Ecology has compelling examples, like competition versus colonization, or longevity vs fecundity. But in your case, I struggle to find the competing processes. In the end, this is just a terminology problem but it may lead to some ecologists to confusion. It would be great if you could clarify the "trading-off" elements, but I would prefer too call it alternative strategies. In any case, I really like Table 1 (except for the mention to trade-off, as you could understand).

Copiotrophs and oligotrophs.

The paragraph going from line 307 to 323 adds confusion to the manuscript. I do not recall mentions of copiotrophs and oligotrophs until this moment in the manuscript, and as you say there is no exact agreement with the traditional view of these terms. So, I think you can skip this entire paragraph, for the sake of clarity and a simpler train-of-thought.

Minor comments.

l. 11 Bacteria can not assess the expected return of an investment. Please delete the phrase and rephrase l. 215 and l. 397 too.

l. 21-25 This sentence is too long and not so clear. Please reformulate, maybe cutting it in half.

l. 28 It is the first time that as a community ecologist I encounter the EC 3.2.1.* nomenclature. The reader might need to see Enzyme Class (EC) before adding directly the abbreviature. Moreover, the example provided is odd, you mention enzymes and give an example of substrates such as cellulose, which is no an enzyme. In this case, I’d be more verbose.

l. 334 This is your third hand. Please rephrase.

l. 324-353 Please discuss whether your diffusion mechanism is similar or not to the actual situation in soils. I would say that in soils there are pulses of diffusion (caused by rain) rather than a constant one.

l. 372-398 I find this paragraph too speculative. Please anchor it better in the literature or just delete it.

l. 399-407 I am sure that you can do a much more convincing conclusion than this one. Why is your manuscript relevant? What are your biological insights? Fly high.

- There are typos in Figure 5 A and C.

- Figures are in general too small and with many panels. Maybe you could consider to send some to the supplementary and make the rest a bit bigger.

- The github code for the first part needs more explanation in the read.me. It should be clear how to run the code in both repositories. Not all ecologists know Python (R is much more common for ecologists).

- Maybe you could discuss how including explicit mechanisms of polymer degradation or consumption of resources changes ecological insight into the dynamics of microbial communities. E.g. https://doi.org/10.1007/s12080-020-00466-7

**Have the authors made all data and (if applicable) computational code underlying the findings in their manuscript fully available?**

Reviewer #1: None

Reviewer #2: Yes

PLOS authors have the option to publish the peer review history of their article (what does this mean?). If published, this will include your full peer review and any attached files.

Reviewer #1: **Yes: **Stefano Manzoni

Reviewer #2: **Yes: **Vicente J. Ontiveros
---

## [Decision Letter · Decision Letter 1]

25 Mar 2024

Dear Dr Guseva,

Thank you very much for submitting your manuscript "Bacteria face trade-offs in the decomposition of complex biopolymers" for consideration at PLOS Computational Biology. As with all papers reviewed by the journal, your manuscript was reviewed by members of the editorial board and by several independent reviewers. The reviewers appreciated the attention to an important topic. Based on the reviews, we are likely to accept this manuscript for publication, providing that you modify the manuscript according to the review recommendations.

Please address the remaining issues pointed out by the reviewers. In particular, Reviewer #2’s suggestions for improving the structure of the Introduction seem very constructive and could make this work more easily accessible to a broad audience.

Sincerely,

Tobias Bollenbach

Academic Editor

PLOS Computational Biology

Natalia Komarova

Section Editor

PLOS Computational Biology

Please address the remaining issues pointed out by the reviewers. In particular, Reviewer #2’s suggestions for improving the structure of the Introduction seem very constructive and could make this work more easily accessible to a broad audience.

Reviewer's Responses to Questions

**Comments to the Authors:**

Reviewer #1: First of all, apologies for the slight delay in submitting this review. The revised manuscript has improved and the model is described more clearly. I have a few remaining minor comments and one technical question. Starting from the latter: could you provide some more explanations regarding Eq. (6) in the Supplementary materials? As written, it does not depend on i, and for any j>1 we get g<1, which is strange since g is defined as the number of molecules of size i (not a fraction). Moreover, j-1 simplifies away when plugging Eq. (6) into Eq. (7).

Main text

L150: not clear what ‘label substrate’ means

L163: N eliminated from the cell would accumulate as mineral N?

L164: a lower limit for cell biomass?

L357-359: the reason why oligomers are lost and not used on site is explained around L405, but it would be useful to explain it here already

Supplementary materials

Several symbols are not included in Table S4; e.g., U_{cell}, n_{col}, C_b, dt, k_1

First sentence at the beginning of Section B: missing words

Second line after Eq. (7): “numerator”

Second line in Subsection (i): “take up only monomers”

One line above Eq. (10): I would delete “as”

Eq. (11): what is the difference between U_{cell} and U_{pot}?

Eq. (13): perhaps it would be simpler to use symbol U_c already here, since uptake = U_c

Second line in Subsection (ii): do you mean “to meet maintenance requirements”?

Second line in Subsection (vi): or rather limited motility, since the colony partly moves to nearby grid cells if it becomes too large

Sixth line in Subsection (vi): other components being soil matrix and organic matter particles?

Eight line in Subsection (vi): I am not sure I understand—when there are three cells, the colony is split in half (1.5 cells?) and one half is moved to nearby grid cells? Or when reaching a number of four cells, so the colony is split in two small colonies of two cells each? Also, this threshold value for colonization of nearby cells seems much smaller than the theoretical maximum of 1000 cells reported above

First line in Section C: “We start with…”

Third line in Section C: “give an initial boost to…”

Second last line of P5: “to guarantee”

Fig. S3: how is the presence of soil matrix accounted for in the definition of D_n? generally diffusivities in pore water are scaled by soil saturation level (or porosity if the soil is saturated)

First line just below the caption of Fig. S3: please add units to the values of dt

Reviewer #2: The authors have improved the manuscript greatly following both reviewers remarks. Most of my comments have been satisfactorily addressed. But we still have some disagreements. I still think that alternative strategies to polymer decomposition describes better than trade-off the situation of endo- and exo-enzymes. Trade-offs usually involve different processes (growth Vs enzyme production or defense strategies), but I agree that the limits are fuzzy, so I do not see a big problem in using the term. As for the introduction, it is still problematic in my opinion (see below). And I have seen some minor details to be addressed, nothing very problematic. In any case, this is a really interesting contribution that advances our understanding of the processes underlying microbial ecology.

Introduction.

I’m not convinced yet that the introduction is as good as it can be. I have no problems with the first two paragraphs. The first conveys that the decomposition of organic matter is complex and needs multiple enzymes with an increased metabolic cost. Moreover, high variability in the system hinders the processing of organic matter. However, we see diversification in the microbial response to tackle this complex environment, which hints at the existence of trade-offs. The second paragraph gives an overview of degradation, explaining endo- and exo-enzymes.

In the third paragraph, you introduce the two literature surveys, to investigate the prevalence of both type of enzymes and if there are specialists or generalists. This will be done using the case study of chitinases. The next paragraph gives the results of the first survey, that there is a plethora of combinations but it is still difficult to assess prevalence. The fifth paragraph indicates that essays show that the most common strategy is to be a generalist, and raises the main questions of the article. Finally, you indicate that there is a gap in knowledge, that has been partially addressed, but that you are going to address it from a more comprehensive perspective.

In this structure, lines 49-64 are hard to follow, with lines 49-51 that are a bit disorienting because they talk in the past about results that have not been introduced yet. Also, l.76-83 feel like an step back to the first paragraphs of the introduction. In summary, I believe the flow of the introduction is not as smooth as it should be, mainly because of the inclusion of the survey results in the introduction.

I think that an alternative structure would solve this, streamlining the introduction and improving flow. The first paragraph may stay the same as it is. First, I would introduce l.76-83 after l.34, where you introduce that endo- and exo-enzymes produce oligomers of diferent sizes. In that way, you add here the gaps in understanding of the depolimerization process and how this has been partially adressed before. Then, I would set up the stage to your way of adressing these gaps: a case study of chitinases. Chitinases would allow you to explore the prevalence of endo- and exo-enzymes and to what extent microbes adopt specialist or generalist roles, your first two questions. Note that you do not need to answer straight away both questions. I think you can reasonably expect that generalists could be the norm based on references 10-13. And that would allow you to introduce this expectation and the questions that you introduce in lines 71-75. And you can add the gaps that you are addressing l.84-98 with some polishing. Of course, this would imply the results of the survey into the results section, and maybe adding some comment in the discussion, but I think you would end up with a rounder and more effective contribution.

Minor details

By the way, you talk about Figure 2D, that is not actually present in the text. I believe that your present references to Figures 2B, C and D should be actually 2A, B and C, because you moved old figure 2A to the supplementary.

You have Enzyme Classes and Enzyme Commission number for the same abbreviation (EC). I believe that the first appearance should be Enzyme Commission class. I got a bit confused when I saw both having the same abbreviation.

Finally, the term copiotrophic applied to environments is unusual, because it implies a specific strategy of the organisms. I think the readers would be more fond of the term carbon-rich environment instead.

**Have the authors made all data and (if applicable) computational code underlying the findings in their manuscript fully available?**

Reviewer #1: Yes

Reviewer #2: Yes

PLOS authors have the option to publish the peer review history of their article (what does this mean?). If published, this will include your full peer review and any attached files.

Reviewer #1: **Yes: **Stefano Manzoni

Reviewer #2: **Yes: **Vicente J. Ontiveros

Figure Files:

Data Requirements:

Reproducibility:

References:

---

## [Decision Letter · Decision Letter 2]

12 Jul 2024

Dear Dr Guseva,

We are pleased to inform you that your manuscript 'Bacteria face trade-offs in the decomposition of complex biopolymers' has been provisionally accepted for publication in PLOS Computational Biology.

Best regards,

Natalia L. Komarova

Section Editor

PLOS Computational Biology

Natalia Komarova

Section Editor

PLOS Computational Biology

Reviewer's Responses to Questions

**Comments to the Authors:**

Reviewer #1: The author addressed my comments and I do not have any further concern.

Reviewer #2: The authors have followed all recommendations and the flow of the manuscript has improved. I have no further comments, and I thank the authors for this great contribution.

**Have the authors made all data and (if applicable) computational code underlying the findings in their manuscript fully available?**

Reviewer #1: None

Reviewer #2: Yes

PLOS authors have the option to publish the peer review history of their article (what does this mean?). If published, this will include your full peer review and any attached files.

Reviewer #1: **Yes: **Stefano Manzoni

Reviewer #2: **Yes: **Vicente J. Ontiveros

---

## [Editor Report · Acceptance letter]

31 Jul 2024

PCOMPBIOL-D-23-01732R2 

Bacteria face trade-offs in the decomposition of complex biopolymers

Dear Dr Guseva,

I am pleased to inform you that your manuscript has been formally accepted for publication in PLOS Computational Biology. Your manuscript is now with our production department and you will be notified of the publication date in due course.

With kind regards,

Anita Estes
